# Electrochemical Detection of Dopamine at a Gold Electrode Modified with a Polypyrrole–Mesoporous Silica Molecular Sieves (MCM-48) Film

**DOI:** 10.3390/ijms20010111

**Published:** 2018-12-29

**Authors:** Izabela Zablocka, Monika Wysocka-Zolopa, Krzysztof Winkler

**Affiliations:** Institute of Chemistry, University of Bialystok, Ciolkowskiego 1K, 15-245 Bialystok, Poland; izasoltys@wp.pl (I.Z.); monia@uwb.edu.pl (M.W.-Z.)

**Keywords:** mesoporous silica, polypyrrole, MCM-48, polypyrrole composite, dopamine

## Abstract

A gold electrode modified with a polypyrrole–mesoporous silica molecular sieves (polypyrrole—MCM-48) nanostructure film was used for the electrochemical determination of small concentrations of dopamine (DA) by cyclic voltammetry and square-wave voltammetry techniques. This electrode showed good electrocatalytic activity for the oxidation of dopamine. The oxidation potential of dopamine was decreased significantly compared with that obtained at the bare gold electrode. The observed linear range for the determination of the dopamine concentration, without interferents through cyclic voltammetry measurements, was from 10 μM to 1.2 mM (R^2^ = 0.9989) for the gold electrode modified with the polypyrrole—MCM-48 nanostructure, with a detection limit of 2.5 μM. In the case of square-wave voltammetry, the linear range was 2–250 μM, with a correlation coefficient of 0.9996, and the detection limit was estimated to be 0.7 μM. The effects of interferents, such as ascorbic acid (AA) and uric acid (UA), on the electrochemical detection of dopamine were also examined. The modified electrode can successfully separate the oxidation potentials for ascorbic acid and dopamine, shifting the oxidation peak potential of ascorbic acid to a more positive potential, and significantly decreasing the peak current. The presence of ascorbic acid increased the sensitivity of dopamine determination at the modified electrode, and the detection limit was estimated to be 0.5 μM with 0.1 mM ascorbic acid to imitate physiological solutions. Additionally, studies showed that the presence of uric acid does not affect the electrochemical detection of dopamine. The modified electrode can be successfully applied for the quantitative analysis of dopamine both with and without interferents.

## 1. Introduction

Dopamine is one of the most important neurotransmitters in the central nervous, renal, and hormonal systems, as well as in drug addiction and Parkinson’s disease [1,2]. Dopamine usually exists in blood serum. Therefore, the determination of the concentration of this neurochemical is important. Dopamine is a type of catecholamine that can be detected electrochemically upon its oxidation on an electrode surface [3]. This electrochemical method has been widely used as a direct measurement technique for dopamine because it has several advantages, such as small dimensions, a fast response, high accuracy, and easy operation. The main problem with electrochemical detection is the interference due to ascorbic acid and uric acid, because they have similar oxidation potentials, close to that of dopamine on the bare working electrode [4]. However, various approaches have been applied to overcome these difficulties. As is well known, modified electrodes may offer higher selectivity, sensitivity, and stability than the bare electrodes. Therefore, several modifier materials such as polymers [5,6,7,8,9,10,11,12], metal nanoparticle arrays [13,14,15,16,17,18], carbon materials [19,20,21,22,23,24], and metal oxides [25,26,27,28] have been used to overcome the problem of interference. For example, Oshaka and co-workers studied the voltammetric behavior of dopamine in the presence of ascorbic acid at an electropolymerized film of N,N-dimethylaniline, coated on a glassy carbon electrode [7]. This electrode showed a stable response without fouling of the electrode surface by the adsorption of the oxidized product of ascorbic acid. Graphene was also used for the electrochemical determination of dopamine [21,22]. It can be used as material for the selective detection of dopamine because ascorbic acid is completely eliminated under these conditions. The voltammetric separation of dopamine and ascorbic acid occurred when a graphite electrode was modified with ultrafine TiO_2_ powder [27]. Ramanavicius and co-workers reported that modification of glassy carbon electrodes with copper nanoparticlesallows for dopamine determination in the presence of ascorbic acid, uric acid, and *p*-acetamidophenol [13]. They observed very high sensitivity of dopamine determination at such an electrode.

Recently, conductive polymers and their composites have been used to modify electrodes to detect dopamine [29,30,31,32,33,34,35,36,37]. Polypyrrole is one of the most important conducting polymers, and it has been an extensively studied material during the last decade because it can be used in a neutral pH region and its stable film can be easily deposited, both chemically and electrochemically, in the form of various structures onto a variety of substrate materials [38,39,40,41]. The formation of the polypyrrole film and nanocomposites with polypyrrole is extremely attractive in the field of electrochemical sensors because these materials increase the surface area of the electrode, enhance conductivity, and facilitate electron transfer [42,43,44,45].

MCM-48 is a cubic form of mesoporous silica molecular sieves with a three-dimensional pore structure. Such a system exhibits high specific surface area, large pore volume, and high thermal stability [46,47,48]. These materials can be combined with conducting polymers to form composites. In the porous silica templates, conducting polymers are deposited in the pores as tubular or fibrous structures. 

In this paper, we report the electrochemical behavior of dopamine at the surface of a polypyrrole—MCM-48 nanostructure-modified Au electrode, both in the presence and absence of ascorbic acid and/or uric acid. The modified electrode shows electrochemical activity for the oxidation of dopamine. The nature of the polypyrrole film is also very important and affects the determination of dopamine. In addition, we discuss the suitability of these modified electrodes in the voltammetric determination of dopamine in a phosphate buffer solution.

## 2. Results and Discussion

### 2.1. Formation and Electrochemical Properties of the Modified Electrodes

The polypyrrole—MCM-48 nanocomposite powder was dispersed by sonication in dichloromethane solution to form a homogenous mixture. Ten microliters of a 1.5 mg/mL mixture was dropped onto the surface of the Au electrode, and the solvent was evaporated. The morphology of the gold electrode surface modified with polypyrrole—MCM-48 nanocomposite is shown in Figure 1. The composite films formed a uniform layer composed of spherical particles of silica. This modified electrode was characterized by cyclic voltammetric measurements. Figure 2 shows the voltammetric behavior of a thin film of the polypyrrole—MCM-48 composite in the 0.1 M phosphate buffer solution. In this case, the material was electrochemically active at positive potentials due to polypyrrole oxidation. This composite exhibited electrochemical reversibility with one oxidation (O_1_) peak and one reduction (R_1_) peak. The morphology of the polypyrrole—MCM-48 composite is shown in the inset of Figure 2. 

### 2.2. Voltammetric Behaviors of Dopamine at the Bare Gold Electrode

The cyclic voltammograms obtained for dopamine oxidation at the bare gold electrode in 0.1 M of phosphate buffer solution are shown in Figure 3. The electrochemical oxidation of dopamine in aqueous solution goes through three steps according to the ECE mechanism presented in Scheme 1, where E and C indicate the electrochemical and chemical steps, respectively [49,50]. In the first scan (Figure 3, curve *1*) dopamine is oxidized to dopaquinone (peak a_1_), and next, it is reduced in part to dopamine (peak b_1_). Part of dopaquinone undergoes a cyclization reaction to form leucodopaminechrome, which is immediately oxidized to dopaminechrome, and next, this dopaminechrome is reduced back to form leucodopaminechrome (peak b_2_). Therefore, in the second scan (Figure 3, curve *2*) an a_2_ peak appears, corresponding to the oxidation of leucodopaminechrome to dopaminechrome, while the dopamine oxidation peak, a_1,_ decreases at the expense of the peak pair, a_2_ and b_2_.

### 2.3. Electrochemical Detection of Dopamine at the Gold Electrode Modified with the Polypyrrole—MCM-48 Nanocomposite

In the case of the polypyrrole—MCM-48-modified electrode, the analyte has to enter the pores of the mesoporous material before electrochemical measurements. Therefore, the effect of the time of preconcentration was examined. Figure 4 shows the cyclic voltammograms recorded for different preconcentration times at the electrode modified with the polypyrrole—MCM-48 nanocomposite in the phosphate buffer solution containing dopamine. The mediated oxidation peak current of dopamine depends on the preconcentration time. Namely, the oxidation peak current rapidly increases with prolonging the preconcentration time of the modified electrode, from one to 10 min in solution, and then, it reaches an almost constant value for a longer time. The optimum current was obtained for a preconcentration time of 10 min (curve *5* in Figure 4). Therefore, for all practical measurements, 10 min of preconcentration time was found to be sufficient for the determination of dopamine.

The influence of the pH of the solution on dopamine oxidation was studied by cyclic voltammetry in the presence of 0.1 mM of dopamine in an aqueous buffer solution, with increasing pH values, from 4.2 to 9.0. The potential of the anodic peak shifted in the negative direction as the pH increased. The linear dependence on pH, with a slope of −0.056 V/pH, indicates that the oxidation of dopamine at the polypyrrole—MCM-48 nanocomposite-modified electrode had been inferred to be a double electron transfer reaction. It was also observed that both the oxidation and reduction peak currents obtained maximum values at a pH of 7.4. Considering the physiological environment, a phosphate buffer solution at a pH of 7.4 was chosen as the supporting electrolyte for the experiments.

Figure 5 presents the cyclic voltammograms recorded at the gold electrode modified with the polypyrrole—MCM-48 nanocomposite (Figure 5a, curve *2*) and, for comparison, on the bare gold electrode (Figure 5a, curve *1*) in 0.1 M of phosphate buffer solution containing 1 mM of dopamine. Under these conditions, the potential of the polypyrrole oxidation peak corresponds to the potential for the oxidation of dopamine. This observation demonstrates that the polypyrrole—MCM-48 nanocomposite has high electrocatalytic activity towards the oxidation of dopamine. This peak current increased almost three-fold compared with the bare gold electrode. The increase in the sensitivity of the analytical determination of dopamine is a clear indication of the enlargement of the active surface area of the electrode modified by the polypyrrole—MCM-48 nanocomposite. The effect of the sweep rate on the anodic peak current of dopamine was also studied. As the sweep rate increased, the oxidation peak current (*I_pa_*) also increased (Figure 5b). The oxidation peak current was directly proportional to the sweep rate over the range of 0.01–0.2 V s^−1^ (inset of Figure 5b). The results showed that the polypyrrole—MCM-48 film was stable and electrochemically active in the phosphate buffer solution, showing a typical surface adsorption behavior.

The influence of the concentration of dopamine at the gold electrode modified with the polypyrrole—MCM-48 nanocomposite in 0.1 M of phosphate buffer solution, using cyclic voltammetry and square-wave voltammetry, is shown in Figure 6 and Figure 7, respectively. These results show that the peak currents of dopamine oxidation and reduction increased with the increase of the dopamine concentration. The dependence of the anodic peak current on the concentration of dopamine for cyclic voltammetry and square-wave voltammetry is shown in Figure 8. In the case of cyclic voltammetry, a linear relationship for the dopamine concentration is observed in the range of 10 μM to 1.2 mM (inset of Figure 8a), and the linear regression equation is y = 9.7984x + 0.6143, with a R^2^ = 0.9989. The limit of detection (LOD) of dopamine determination was calculated from the following relationship [51]:
LOD = 3s/m(1)
where s is the standard deviation of the peak current of the blank signal and m is the slope of the calibration plot. The LOD was found to be 2.5 μM. The plot of the peak current as a function of the concentration for square-wave voltammetry gave a linear line in the range from 2 μM to 0.25 mM (inset of Figure 8b), and the linear regression equation was y = 0.0058x + 0.0383, with R^2^ = 0.9996. The limit of detection was determined to be 0.7 μM. Thus, the prepared electrode allows for the sensitive determination of dopamine in a relatively large concentration range. The comparison of the results for the determination of dopamine using different polypyrrole composite-modified electrodes is summarized in Table 1. For the method based on the square-wave voltammetry, the linear peak current, concentration range, and detection limit were comparable or better in comparison to the electrochemical procedures of dopamine determination at the electrodes modified with polypyrrole-containing composites presented in the literature. Only Qian and co-workers reported better sensitivity of dopamine determination at an electrode modified with core-shell microspheres of polypyrrole and reduced graphite oxide [32]. The gold electrode modified with polypyrrole—MCM-48 nanocomposite shows comparable analytical performance of dopamine determination to electrodes modified with carbon nanomaterials [20] and metallic nanoparticles [14,15,16,17,18].

### 2.4. Interference by Ascorbic Acid and Uric Acid

An important property of the composite that is useful for the evaluation of high-performance sensors is its selectivity, which is a measurement of the ability to reject major interferents, such as ascorbic acid and uric acid. These compounds are presented in physiological solutions at concentrations of 0.1 mM for ascorbic acid and 0.05 mM for uric acid [57,58]. Additionally, the basal dopamine concentration is very low (0.01–1 μM), and thus, the concentration of ascorbic acid is generally much higher [58]. 

The cyclic voltammetric responses of dopamine, uric acid, and ascorbic acid recorded in 0.1 M of phosphate buffer solution at the bare gold electrode and the polypyrrole—MCM-48-modified gold electrode are shown in Figure 9. At the bare electrode (Figure 9a), dopamine and ascorbic acid showed oxidation peaks at approximately 0.25 and 0.3 V, respectively. As expected, it was very difficult to distinguish between them because their oxidation peak potentials were very close. However, uric acid showed irreversible oxidation peaks at approximately 0.5 V. Under these conditions, the oxidation peaks of dopamine and uric acid were very well separated, and the peaks of ascorbic acid and dopamine were not separated. 

In the case of the gold electrode modified with the polypyrrole—MCM-48 (Figure 9b), the potentials of dopamine and uric acid oxidation were very similar to those observed at the bare gold electrode. However, the oxidation potential of ascorbic acid was shifted to more a positive potential, by 200 mV, in comparison to the gold electrode. The shift in the peak potential was due to a kinetic effect [59]. Therefore, both uric acid and ascorbic should not disturb the detection of dopamine, as the oxidation peak for ascorbic acid shifted significantly toward a more positive potential, thus separating it from the oxidative peak of dopamine. Furthermore, the peak current of dopamine increased almost three-fold, but the peak current of ascorbic acid decreased almost five-fold compared to those on the bare electrode. The repulsion of negatively-charged ascorbic acid from the electrode due to the negatively-charged layer of nanocomposite was responsible for shifting its electrochemical oxidation to a more positive potential. The modified electrode could also inhibit the redox current of ascorbic acid, significantly reducing the current response. As the difference in the oxidation peak potentials for ascorbic acid and dopamine at the modified electrode was approximately 0.2 V, it can be expected that this electrode could separate the oxidation of dopamine and ascorbic acid, coexisting in the same solution, well.

### 2.5. The Influence of Ascorbic Acid or Uric Acid on the Determination of Dopamine

The influence of ascorbic acid and uric acid on the determination of dopamine was examined using both cyclic and square-wave voltammetry. Figure 10 presents the effect of the ascorbic acid concentration on the process of dopamine oxidation at the gold electrode modified with the polypyrrole—MCM-48. Ascorbic acid influences the dopamine oxidation process only when the concentration of ascorbic acid is 40 times higher than the concentration of dopamine. The fact that the concentration of ascorbic acid is one order of magnitude higher than dopamine, in the biological environment, results in high selectivity for the detection of dopamine at this modified electrode. When this ratio of the ascorbic acid to dopamine concentration was smaller, only the anodic peak for the oxidation of dopamine was observed, but the presence of ascorbic acid increased the sensitivity of dopamine determination. The detection limit for dopamine using square-wave voltammetry was 0.5 μM, in a solution containing 0.1 mM of ascorbic acid to imitate physiological solution, while without ascorbic acid, the limit of detection was 0.7 μM. The fact that the coexistence of ascorbic acid in the measurement solution notably enhanced the current signal for dopamine oxidation is probably due to ascorbic acid’s ability to reduce dopaquinone to dopamine, thus amplifying the oxidation current of dopamine at the surface electrode. This effect results in the improvement of the sensitivity of dopamine determination [60].

The electrooxidation of dopamine in the presence of uric acid under similar conditions was also studied. Typical cyclic and square-wave voltammograms are shown in Figure 11. The oxidation peaks were obtained at approximately 0.17 V for dopamine and 0.55 V for uric acid using the electrode modified with the polypyrrole—MCM-48. As can be seen for the mixture of dopamine and uric acid, the oxidation peak for dopamine was shifted towards a more negative potential than in the solution containing only dopamine. Therefore, better peak separation was observed. At the electrode modified with polypyrrole—MCM-48, the presence of uric acid did not affect the electrochemical detection of dopamine.

### 2.6. Simultaneous Detection of Dopamine, Ascorbic Acid, and Uric Acid

The simultaneous determination of the mixture of dopamine, ascorbic acid, and uric acids at the bare gold electrode and modified electrode were studied by cyclic voltammetry. Figure 12 shows the cyclic voltammetry responses for the solution containing relatively high concentrations of ascorbic acid, uric acid, and dopamine, recorded at the bare electrode and the electrode modified with the polypyrrole—MCM-48 nanocomposite. Voltammograms recorded at the bare electrode showed two peaks (curve *1* in Figure 12). The oxidation peaks of dopamine and ascorbic acid completely overlapped at 0.4 V, making the simultaneous determination of these compounds impossible. The oxidation peak of uric acid appearred at about 0.55 V. Curve *2* in Figure 12 was obtained in the mixture of dopamine and a high concentration of ascorbic acid at the modified electrode with polypyrrole—MCM-48 film. Cyclic voltammograms recorded in the solution containing a mixture of dopamine, ascorbic acid, and uric acid at the modified electrode are shown on the curves *3*, *4*, and *5* in Figure 12, for the same concentration of dopamine and ascorbic acid, and three different concentration of uric acid. In this case, the oxidation currents of ascorbic and uric acids overlapped, leading to the formation of a large oxidation peak current at 0.55 V. The peak of dopamine oxidation was separated from the peak current corresponding to ascorbic acid and uric acid oxidation, by ca. 250 mV, and appeared at 0.30 V on the cyclic voltammograms. A slight shift of dopamine oxidation peak potential, toward more positive values, was observed with the increase of uric acid concentration in the solution.

## 3. Conclusions

The electrochemical behavior of the polypyrrole—MCM-48 nanocomposite was studied in the presence of dopamine, both without and with interferents such as ascorbic acid and uric acid. The experimental results show that the oxidation of dopamine is catalyzed at the surface of the modified gold electrode. These materials have been also used to compare their abilities to detect small concentrations of dopamine, both without and with interferents. The larger oxidation peak separations, between dopamine, ascorbic acid, and uric acid, on the modified gold electrode make it suitable for the determination of dopamine in the presence interferents. The modified electrode showed good selectivity and high sensitivity, even in the presence of interferents.

## 4. Experimental Section

### 4.1. Materials

Dopamine, ascorbic acid, and uric acid were purchased from Aldrich and were used as received. The 0.1 M phosphate buffer solution with a pH of 7.4 was prepared with KH_2_PO_4_ and Na_2_HPO_4_. Freshly prepared solutions of dopamine were used in all experiments. Pyrrole and FeCl_3_ were used as received from Aldrich Inc. The *n*-hexadecyltrimethylammonium bromide template, tetraethyl orthosilicate, aqueous ammonia, and ethanol were used as received from the Aldrich Chemical Co. These were used for the synthesis of MCM-48 spheres. Pure deionized water, with a resistivity of 18.2 MΩ cm, was obtained from a Milli-Q Millipore system.

### 4.2. Instrumentation

Voltammetric experiments were performed on an AUTOLAB Model 283 Potentiostat/Galvanostat (EG&G Princeton Applied Research, Oak Ridge, TN, USA) with a three-electrode cell. The AUTOLAB system was controlled with GPES 4.9 software by the same manufacturer. A gold disk electrode with a diameter of 1.5 mm (Bioanalytical Systems Inc., West Lafayette, IN, USA) was used as the working electrode. Prior to each experiment, the electrode was washed with acetone and dried. An Ag/AgCl/saturated KCl electrode was used as the reference electrode. The counter electrode was a platinum tab with an area of approximately 0.5 cm^2^.

Scanning electron microscopy (SEM) images were obtained using an Inspect S50 microscope (FEI Company, Hillsboro, OR, USA). The accelerating voltage of the electron beam was either 20 or 25 kV and the average working distance was 10 mm.

Transmission electron microscopy (TEM) images were recorded with a FEI Tecnai TF20 (Thermo Fisher Scientific, Waltham, MA, USA) instrument using an accelerating voltage of 200 kV. The films for the TEM studies were prepared by constant potential electrodeposition on a gold grid with a diameter of 3 mm (300 square mesh).

### 4.3. Preparation of the Polypyrrole—MCM-48 Nanocomposite

The mesoporous silica MCM-48 was synthesized according to the procedure proposed by Schumacher et al. [46]. This synthesis leads to the formation of spherical particles with diameters in the range of 400 to 800 nm. Polypyrrole was deposited in the mesopores of the MCM-48 host by pyrrole wetting due to the capillary effect, followed by chemical polymerization in the presence of FeCl_3_ [47]. For this purpose, first MCM-48 and 0.1 M pyrrole were added to a solution of dichloromethane and stirred for 2 h at room temperature. After this time, the precipitate was washed with pure dichloromethane 4 times to remove the pyrrole from the surface of the silica and dried at room temperature. In this manner, pyrrole-filled mesopores of MCM-48 were obtained. Next, this product was dissolved in deionized water, and FeCl_3_ was added and stirred for 2 h at room temperature. The composite of polypyrrole@MCM-48 was washed with pure water and methanol 4 times and dried at 80 °C. The color of the silica changed from white for pure MCM-48 to beige for the composite. Morphological parameters of these materials were obtained by nitrogen sorption measurements [47]. Both samples, pure MCM-48 and polypyrrole@MCM-48 composite, showed a BET surface of 1520 m g^−1^ and 1270 m g^−1^, respectively. The incorporation of polypyrrole into the pores of MCM-48 resulted in a decrease in the average pore diameter, from 3.2 nm for MCM-48 to 2.7 nm for the composite. These changes in the pore diameter corresponded to changes in the pore volume, from 0.74 to 0.70 cm^3^ g^−1^, respectively. The small changes of pore volume indicated that polymer only partially fills the MCM-48 pores. Namely, this polymer is deposited only on the walls of mesopores [47].

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
