# Peer review of "Electrochemical Detection of Dopamine at a Gold Electrode Modified with a Polypyrrole–Mesoporous Silica Molecular Sieves (MCM-48) Film"

_ijms, 2018, doi:10.3390/ijms20010111_

Round 1

Reviewer 1 Report

Reviewer report on Manuscript Draft entitled ‘Electrochemical detection of dopamine at a gold electrode modified with a polypyrrole@MCM-48 film’

In this research a gold electrode modified with a polypyrrole@(Mesoporous molecular sieves) nanostructure film was used for the electrochemical determination of small concentrations of dopamine by cyclic voltammetry and square-wave voltammetry techniques. Manuscript is interesting from electrochemical and electroanalytical points of view. Therefore, eventually it can be published after some minor improvements and some corrections:

All abbreviations should be clearly introduced before usage of them in abstract and elsewhere in the text, e.g. abbreviation MCM-48 – Mesoporous molecular sieves is not introduced.

I will recommend to remove the abbreviation from the Title instead of this I will recommend to use full title ‘Mesoporous molecular sieves‘.

Figure 2. it will be nice to present CV responses during different stages of the formation Ppy-based layer.

References on electrochemical determination of dopamine (Copper nanoparticle modified carbon electrode for determination of dopamine Electrochimica Acta 2012, 76, 201–207.) should be better overviewed, compared and discussed in corresponding parts of the manuscript.

Figure 8. error bars indicating standard deviation should be presented, in the case if experiment was performed for several times.

English need some improvements.

Author Response

Reply to Reviewer #1 comments:

All abbreviations should be clearly introduced before usage of them in abstract and elsewhere in the text, e.g. abbreviation MCM-48 – Mesoporous molecular sieves is not introduced.

Abbreviation MCM-48 has been introduced in the abstract and in the ‘Introduction’ part of the manuscript.

I will recommend to remove the abbreviation from the Title instead of this I will recommend to use full title ‘Mesoporous molecular sieves‘.

The title of the manuscript was changed according to Reviewer’s comment.

Figure 2. it will be nice to present CV responses during different stages of the formation Ppy-based layer.

There is some misunderstanding of the concept of modified electrode formation reported in this work. The composite material has been formed by chemical (not electrochemical) polymerization of pyrrole in the pores of silica. In the next step, the composite has been deposited at the electrode surface by drop coating method.

References on electrochemical determination of dopamine (Copper nanoparticle modified carbon electrode for determination of dopamine Electrochimica Acta 2012, 76, 201–207.) should be better overviewed, compared and discussed in corresponding parts of the manuscript.

The different aspects of electrochemical determination of dopamine have been additionally discussed in the ‘Introduction’ part of the paper (page 2) particularly in terms of dopamine determination in the presence of typical interferences. New references have been added including the reference suggested by Reviewer. On page 11, the analytical performance of electrode modified with polypyrrole@MCM-48 in  dopamine determination is also  compared to the other systems (carbon nanomaterials, metallic nanoparticles) reported in literature

Figure 8. error bars indicating standard deviation should be presented, in the case if experiment was performed for several times.

Figure 8 has been corrected.

English need some improvements.

Before submission to the Editor of the journal, the manuscript was corrected by Elsevier Language Editing Service. The certificate is shown below.

Reviewer 2 Report

This is a very comprehensive study, the author should address following comments during the revision

1. Scale bar in Figure 2 is very hard to read

2. Many figure insets are also very hard to read, such as in Figure 8

3. The authors need to show the characterization of MCM-48, including pore size and nanoscale morphology. Some previous papers can be reference in how to understand these porous materials Microporous and Mesoporous Materials 2016, 227, 57-64

Author Response

Reply to Reviewer #2 comments:

…the author should address following comments during the revision

1. Scale bar in Figure 2 is very hard to read

Figure 2 and other figures (comment 2) have been corrected.

2. Many figure insets are also very hard to read, such as in Figure 8

3. The authors need to show the characterization of MCM-48, including pore size and nanoscale morphology. Some previous papers can be reference in how to understand these porous materials Microporous and Mesoporous Materials 2016, 227, 57-64

The morphological parameters of MCM-48 and polypyrrole@MCM-48 composite have been reported in our previous paper (ref. [47]). In this manuscript, they have been briefly summarized on page 4.